# A Comprehensive Study on Starch Nanoparticle Potential as a Reinforcing Material in Bioplastic

**DOI:** 10.3390/polym14224875

**Published:** 2022-11-12

**Authors:** Herlina Marta, Claudia Wijaya, Nandi Sukri, Yana Cahyana, Masita Mohammad

**Affiliations:** 1Department of Food Technology, Faculty of Agro-Industrial Technology, Universitas Padjadjaran, Bandung 45363, Indonesia; 2Research Collaboration Center for Biomass and Biorefinery between BRIN and Universitas Padjadjaran, Bandung 45363, Indonesia; 3Solar Energy Research Institute, Universiti Kebangsaan Malaysia, Bangi 43600, Selangor, Malaysia

**Keywords:** starch, starch nanoparticle, reinforcing material, bioplastic, biodegradable, characteristics

## Abstract

Starch can be found in the stems, roots, fruits, and seeds of plants such as sweet potato, cassava, corn, potato, and many more. In addition to its original form, starch can be modified by reducing its size. Starch nanoparticles have a small size and large active surface area, making them suitable for use as fillers or as a reinforcing material in bioplastics. The aim of reinforcing material is to improve the characteristics of bioplastics. This literature study aims to provide in-depth information on the potential use of starch nanoparticles as a reinforcing material in bioplastic packaging. This study also reviews starch size reduction methods including acid hydrolysis, nanoprecipitation, milling, and others; characteristics of the nano-starch particle; and methods to produce bioplastic and its characteristics. The use of starch nanoparticles as a reinforcing material can increase tensile strength, reduce water vapor and oxygen permeability, and increase the biodegradability of bioplastics. However, the use of starch nanoparticles as a reinforcing material for bioplastic packaging still encounters obstacles in its commercialization efforts, due to high production costs and ineffectiveness.

## 1. Introduction

Conventional plastic made from petrochemicals has taken on significant importance in modern life [1,2,3,4,5]. The increasing use of plastic causes an increase in the amount of plastic waste, which has a bad impact, especially on the environment and ecosystem [5]. One alternative is to substitute plastic packagings for more environmentally friendly options, such as bioplastics [6,7,8] or green composites [9]. Bioplastic refers to plastics that are made from organic or bio-based materials, have the characteristics of being naturally biodegradable or biodegradable, are made of organic materials, or can be degraded naturally [10,11]. The bioplastic degradation process is carried out by natural microorganisms, such as bacteria, fungi, and algae. This process produces CO_2_ gas, methane gas, water, and biomass without any toxic by-products [12,13].

However, in general, bioplastic products are brittle, have low melting temperatures, and have low mechanical strength [1,14,15]. Bioplastics are also prone to disintegration during storage or use, are not chemically resistant, and have high water and oxygen permeability [15,16]. Using reinforcing material or filler is one option to improve the properties of the bioplastic matrix. The purpose of reinforcing material is to improve the properties of the matrix, such as enhancing degradability [1,17], increasing mechanical properties [18], and decreasing oxygen and water vapor permeability [13,19].

The basic material commonly used for the manufacture of reinforcing materials in bioplastics is starch. Starch is a polysaccharide derivative compound in the form of granules and is used by plants as an energy source. Starch granules have a semi-crystalline structure and are composed of amylose and amylopectin, which are complex carbohydrate polymers [20,21,22]. Amylose has a straight chain, while amylopectin has a branched chain [23]. In its use, starch can be used in its original form or can be modified beforehand for better characteristics, or as desired [24,25]. In its application, starch with high amylose content will produce products with higher water absorption [26,27]. Starch with high amylose content will also produce a stiffer gel structure. The ratio of amylose and amylopectin affects texture, solubility, viscosity, stability, gelatinization, moisture retention, retrogradation, and syneresis. These criteria affect the application and characteristics of the resulting product [28,29]. Starch is often modified to increase its functional properties and application [30,31,32].

One starch modification method is to reduce the size of starch particles to nanosize, which can be referred to as starch nanoparticles. Starch nanoparticles have at least 1 particle with a size of less than 300 nm and have a high active surface area [33]. Starch nanoparticles are spherical particles of nanosize that have varying crystallinity, and can also be completely amorphous [34]. In addition to the form of starch nanoparticles, there are also starch nanocrystals. Starch nanocrystals can be interpreted as a crystalline plate resulting from the removal of amorphous regions by hydrolysis. Starch nanocrystals have higher crystallinity when compared to starch nanoparticles [35].

Nowadays, the interest in research on bioplastics and reinforcing materials to produce bioplastics with better characteristics has increased. Starch is a promising resource because it is cheap, biodegradable, and abundant; thus, starch is promising in bioplastics and in reinforcing material production. This literature study is expected to provide in-depth information on the potential use of starch nanoparticles as reinforcing materials in bioplastics.

## 2. Preparation of Starch Nanoparticles for Bioplastic Reinforcing Material

There are two types of starch nanoparticle manufacturing methods, namely, top-down and bottom-up [36,37]. The top-down method focuses on reducing the granule size from bulk to micro- to nanosize, for example, by acid hydrolysis, ultrasonication, homogenization, gamma irradiation, and many more [38,39,40,41]. The bottom-up method focuses on the formation of nano-sized starch molecules from atoms under controlled conditions according to the laws of thermodynamics, such as nanoprecipitation and self-assembly methods [42,43,44]. Top-down methods are simple, require less cost, and are suitable for both laboratory and industrial scales. However, top-down methods are less effective for sources that have an irregular shape and an extremely small size. Bottom-up methods demand more money but require short production time and can generate less destructed starch [45]. Figure 1 illustrates starch nanoparticle production method with top-down and bottom-up approaches.

### 2.1. Physical Method

Examples of treatments that include physical methods are milling [46,47], high-pressure homogenization [48,49,50], ultrasonication [51,52], and gamma irradiation [53,54]. The advantage of the physical method is that there is no use of chemicals or solvents, making it more environmentally friendly and simple. The time required in this method is also relatively shorter. The disadvantage of mechanical treatment is that it can cause damage to the crystalline structure of starch granules.

The physical method tends to result in starch nanoparticles with low crystallinity because the use of energy is very large, so the crystalline structure becomes weak [55]. This does not apply to gamma irradiation, as fragmentation by free radicals that occurs through gamma irradiation happens in the amorphous region, resulting in starch nanoparticles with relatively high crystallinity. Although gamma irradiation uses free radicals to destroy the chemical bonds of starch granules and starch particles become smaller, the free radicals here are not dangerous. Because they are easily soluble in water, the resulting starch nanoparticles do not contain radical components [54]. On the other hand, other methods attack the crystalline region, thus resulting in starch nanoparticles with lower crystallinity.

Among the physical method for starch nanoparticle production, milling produces a larger size of starch nanoparticle. During milling, the reduction in starch granules to nanosize is obtained from the weight and speed of the grinding ball that continues to move and rotate. This movement will produce kinetic energy. This energy will be channeled to the sample, resulting in a reduction in the size sample [56]. It can be seen that corn starch processed by milling produces starch nanoparticles with a size of 245 nm [47], which is in line with another study [57].

The ultrasonication method is considered effective in reducing the size of starch. During ultrasonication, energy is transferred to the starch through a cavitation process, which is a condition where the microbubbles burst and spread in the solution suspension in the form of sound waves. This process involves high speed and generates shear forces that can break the covalent bonds of starch; thus, the starch particle size decreases [58]. Several studies have reported that the ultrasonication method produced cassava starch nanoparticles (35–65 nm) [59] and waxy maize starch nanoparticles (37 nm) [60], whereas high-pressure homogenization can produce sago starch nanoparticles (28.514 nm) [61]. The high pressures produce shear forces that can break starch hydrogen bonds, resulting in a reduction in the size of starch into nanoparticles [62].

### 2.2. Chemical and Enzymatic Method

Examples of chemical production methods for starch nanoparticles are acid hydrolysis [36,63,64] and nanoprecipitation [65,66,67]. The enzymatic method can be performed through enzymatic hydrolysis [68,69,70]. Acid and enzymatic hydrolysis use the same mechanism, which is that the area that will be attacked first is the amorphous region, then the acid and enzyme will attack the more crystalline region at a slower rate [17,71,72]. In nanoprecipitation, the starch is gelatinized first, then precipitated using an antisolvent such as ethanol. At the time of precipitation, starch chains will be broken, resulting in starch with a smaller size [73].

Chemical and enzymatic methods mostly result in starch nanoparticles with high crystallinity [74,75]. The difference between acid and enzymatic hydrolysis is that acid hydrolysis uses acid as a hydrolyzing agent, such as sulfuric acid (H_2_SO_4_) or chloride acid (HCl) [76], while enzymatic hydrolysis uses hydrolase enzyme, such as pullulanase [77]. Although acid hydrolysis can produce high crystallinity starch nanoparticles, the time required for the production is far longer than enzymatic hydrolysis and nanoprecipitation. Most researchers that use acid hydrolysis require at least 5 days for hydrolysis [17,63,78], while enzymatic hydrolysis and nanoprecipitation require less than 1 day [43,70,79,80,81,82]. Potato starch nanoparticles which produce using hydrolysis have a higher crystallinity compared to enzymatic hydrolysis. Potato starch nanoparticles obtained by the enzymatic hydrolysis method have a crystallinity of 13.2% [80]. Meanwhile, potato starch nanoparticles obtained by acid hydrolysis and nanoprecipitation have a crystallinity of 42.2% and 44.1%, respectively [83].

Enzymatic hydrolysis tends to produce a higher yield of starch nanoparticle. A study showed potato starch nanoparticles with a manufacturing method using the pullulanase enzyme on elephant foot yam tuber starch, where the extraction yield obtained was 61.33% [84]. Another study also showed that the use of pullulanase enzymes in corn starch hydrolysis produced starch nanoparticles with a yield of 85% [79], while amadumbe starch nanoparticle produced through acid hydrolysis has a yield of 25% [85]. The differences in yield may be attributed to starch type in addition to hydrolysis type.

Regarding size, acid hydrolysis can produce starch nanoparticles with smaller sizes compared to nanoprecipitation. Putro et al. [86] have reported that the acid hydrolysis method produces potato starch nanoparticle with a smaller granule size than the nanoprecipitation method: 85.1 nm and 165.31 nm, respectively. This might be due to the native starch, with microparticle size being fragmented into smaller particles during acid hydrolysis; some of the amorphous parts were hydrolyzed into simple sugars, resulting in the formation of smaller nanoparticles. A previous study has reported that the granule size of starch nanoparticles also depends on amylose content and starch type; the smallest observed size of starch nanoparticles of potato, waxy maize, sweet potato, pea, tapioca, corn, and sweetcorn ranged from 15 to 50 nm, and the largest size observed was between 80 and 225 nm [82]. However, the size and the yield of the starch nanoparticles are influenced by the amylose content and starch crystallinity, where the starch with the low amylose content will produce starch nanoparticles with small size [73]. Waxy maize starch nanoparticles (1% amylose) produced through acid hydrolysis have a size of 47 nm, while high-amylose corn starch nanoparticles (70% amylose) produced with the same treatment have a size of 118 nm [87].

### 2.3. Combined Method

A combined method can be performed using two or more methods at one time of production. It is possible to combine chemical methods and chemical methods, physical methods, and physical methods, as well as chemical methods and physical methods. Examples of combined methods are the use of acid hydrolysis and ultrasonication, acid hydrolysis and precipitation, enzymatic hydrolysis and acid hydrolysis, and many more [88,89,90,91,92].

The use of the acid hydrolysis method with other methods, such as enzymatic hydrolysis, ultrasonication, and milling as a pre-treatment, can overcome the shortcomings of the acid hydrolysis method, namely, the long treatment time [83]. Such pre-treatment can increase the rate of hydrolysis because it means that the acid can reach and damage the amorphous region more quickly [73]. The study conducted by Zhou et al. [88] showed that the use of acid hydrolysis (hydrochloric acid vapor) combined with ultrasonication produces 80.5% waxy maize starch nanoparticles with the hydrolysis time of 1 h, whereas it usually takes more than 5 days for acid hydrolysis. Mango kernel nanoparticle starch produced by combining these two methods has a size of 24.4 nm with a yield of 24.4% [93]. Using the same method, procedure, and starch source, another study reported that the size of the mango kernel starch nanoparticle granule was 79 nm with a yield of 31.7% [94].

The use of ultrasonication before acid hydrolysis treatment can help acid molecules to reach the starch surface more quickly so that the required hydrolysis time is shorter [95]. The cavitation process that occurs during ultrasonication produces a shear force that has an impact on the damage to the starch surface. The extraction yield obtained is also higher with high crystallinity [35]. The production of starch nanoparticles using several methods is presented in Table 1.

## 3. Properties of Starch Nanoparticles

### 3.1. Amylose Content

Amylose content in starch can be analyzed using the iodine-binding method. When amylose meets iodine, it produces a blue color, while amylopectin reacts with iodine to produce a red–purple color that is not too bright [102]. Minakawa et al. [59] reported that yam starch nanoparticles produced by ultrasonication have higher blue value than corn and cassava starch nanoparticles produced using the same method. Higher blue value indicates a higher amylose content in starch.

Potato starch nanoparticles obtained through the nanoprecipitation method had the highest amylose content (21.39%) compared to the acid hydrolysis method (4.79%) and the combined acid hydrolysis method with ultrasonication (3.08%) [83]. In starch nanoparticles from the acid hydrolysis method, amylose content decreased. This is because the amorphous areas in starch granules are very sensitive to acid, so the amorphous areas can be hydrolyzed and leave crystalline regions [102]. Starch crystallinity also has a relation with amylose and amylopectin. Starch crystallinity is attributed to the packing of double helixes formed by amylopectin side chains; thus, starch with higher amylopectin tends to have higher relative crystallinity [87].

Furthermore, LeCorre et al. [87] have reported that amylose content plays a role in determining the particle size of starch nanoparticles, where starch with low amylose content tends to produce starch nanoparticles with smaller particle sizes. This is because amylose is considered to be able to inhibit the hydrolysis process so that hydrolysis becomes slower and the resulting particle size becomes larger.

### 3.2. Granule Morphology Shape and Size

Granule morphology and the size of starch nanoparticles were analyzed using SEM (scanning electron microscopy), TEM (transmission electron microscopy), or FE-SEM (field-emission scanning electron microscopy) [95]. Granule morphology shape and the sizes of starch nanoparticles from various starch sources and production methods are presented in Table 2. A previous study by Hernández-Jaimes et al. [103] reported that acid hydrolysis treatment caused changes in the shape and size of native starch granules and presented erosions and fractures on the granule surface. The longer the hydrolysis process, the higher the deformation on the surface. As the hydrolysis time increases, deeper surface erosion occurs, which eventually leads to the fragmentation of some starch granules. The presence of some small remnant granules after 15 days was observed. The number of nanoparticles increases with longer acid hydrolysis times.

Furthermore, Minakawa et al. [59] reported that ultrasonic treatment causes granule surfaces to crack and erode progressively, as observed by SEM. Some smaller particles are released with different morphology than their native starch granule. This result was in agreement with another study [60], which reported that the ultrasonication method reduces the granule size of waxy maize starch from 2–15 µm to 20–200 nm. The surface of the starch granules appears to be broken down and eroded progressively with increasing ultrasonication time.

### 3.3. Crystallinity

Crystallinity can be interpreted as a comparison between the mass of the crystalline region with the mass of starch nanoparticles as a whole [73]. Starch crystalline structure can be observed using XRD (X-ray diffractograms). The crystalline type of starch nanoparticle can be influenced by the variety or cultivars of the starch source. Banana starch from different cultivars shows different types of crystalline [108].

Starch crystallinity has a relationship with the double-helix structure of the amylopectin chain, so starches with high amylopectin content tend to have high crystallinity compared to starches with high amylose content [87]. Starch nanoparticles of waxy maize with the acid hydrolysis method have a crystallinity of 69%. Meanwhile, the ultrasonication production method obtained 0% crystallinity [34]. Tapioca starch has a crystallinity of 25.12%, which decreased to 12.53% after the nanoprecipitation process and decreased to 6.49% after the nanoprecipitation + ultrasonication process [81]. This is due to the ultrasonication and nanoprecipitation processes attacking the crystalline region; thus, these methods tend to produce starch nanoparticles with low crystallinity [59,81]. Meanwhile, acid and enzymatic hydrolysis attack the amorphous region first; hence, the starch nanoparticles produced from acid and enzymatic hydrolysis have higher crystallinity [79,96,103]. Plantain starch nanoparticles obtained from acid hydrolysis have a relative crystallinity of 90% [103]. The relative crystallinity and crystallinity patterns of starch nanoparticles from various starch sources and production methods are shown in Table 3.

### 3.4. Thermal Properties

Thermal characteristics can be analyzed using DSC (differential scanning calorimetry) and TGA (thermogravimetric analysis). The parameters observed include T_o_ (onset temperature), T_c_ (endset temperature), T_p_ (peak temperature), T_c_−T_o_ (melting temperature range), and ∆H (enthalpy change) [70]. Starch nanoparticles have lower thermal stability when compared to native starch. This is due to a decrease in the molecular weight of the starch nanoparticles [79]. The ultrasonication and nanoprecipitation processes can reduce the thermal stability of starch nanoparticles compared to native starch. Native tapioca starch has T_o_, T_p_, T_c_, and ∆H values of 57.07 °C, 66.33 °C, 73.73 °C, and 9.84 J/g, respectively. The nanoprecipitation + ultrasonication process reduces T_o_, T_p_, T_c_, and ∆H by 40.77 °C, 50.39 °C, 63.8 °C, and 1.99 J/g, respectively [81]. A lower enthalpy indicates a poorer starch molecular structure [29]. The thermal properties of starch nanoparticles from various starch sources and production methods are presented in Table 4.

## 4. Definition and Preparation of Bioplastic with Starch Nanoparticles as Reinforcing Material

Bioplastic refers to plastics that are made from biomass or are bio-based, plastics that have the characteristics of being biodegradable, or plastics made from biomass that can be degraded naturally [10,110]. Examples of bio-based bioplastic packaging are bio-PE and bio-PET [111]; examples of biodegradable bioplastic packaging are PBS (polybutylene succinate), PCL (polycaprolactone), and PVA (polyvinyl alcohol); examples of bioplastics that have bio-based and biodegradable properties are PLA (polylactic acid), PHA (polyhydroxyalkanoate), and packaging with a mixture of starch, cellulose, or lignin [112,113,114,115,116,117].

The methods commonly used for the manufacture of bioplastic packaging are solution casting [98,118,119,120], injection molding [121,122], compression molding [123], and extrusion [124,125,126]. The casting method is very simple and inexpensive; thus, this method is suitable for use on a laboratory scale [127]. On the other hand, it also has drawbacks because it is not suitable for industrial scale [23]. In the manufacture of bioplastic packaging with a starch matrix, starch gelatinization is carried out first with or without a plasticizer. Subsequently, the mixture of starch and plasticizer was cooled to inhibit the gelatinization of the starch nanoparticle. The film solution is then poured into a mold and heated at a certain temperature and time. After cooling and molding, the film can be removed from the mold [98].

The injection molding method produces packaging with good characteristics with effective production costs, including in large-scale production. Before printing, the dough must first be made which is a mixture of matrix and plasticizer. The injection process is carried out using a piston injection molding machine [128]. The extrusion method is suitable for industrial-scale production and requires low production costs. However, the use of nanoparticles in the extrusion method can increase the tendency of nanoparticles to form aggregates; thus, extrusion may not be suitable for bioplastic manufacturing with a starch nanoparticle as the reinforcing material [16]. The steps in the production of bioplastic packaging using the extrusion method consist of mixing, extrusion, and die-cutting stages to obtain bioplastic samples with certain dimensions. The extrusion process is carried out using an extruder machine [124].

## 5. Characteristics of Bioplastic with Starch Nanoparticles as Reinforcing Material

This paper will discuss the mechanical properties, water vapor permeability, thermal properties, and biodegradability of bioplastic with various starch nanoparticles for reinforcement. All the literature used in discussing the characteristics below use casting as a bioplastic production method. A lot of researchers use the casting method because the casting method is the simplest and most inexpensive method, compared to other methods. The characteristics of bioplastic with starch nanoparticle reinforcement will be influenced by filler concentration, filler–matrix interaction, and filler characteristics [129,130].

### 5.1. Mechanical Properties

Parameters tested include flexibility, strength, and stiffness of the bioplastic film. Mechanical characteristics can be analyzed using a TA (texture analyzer) [101] or using DMA (dynamic mechanical analysis) [131]. The effect of the use of reinforcing material on the strength of bioplastic packaging might be caused by the bond between the surface of the reinforcing material and the matrix, resulting in a stress distribution through the friction mechanism between the reinforcing material and the matrix to increase the strength of the bioplastic packaging [85]. The use of reinforcing materials causes the packaging structure to become denser and more compact, thus increasing the mechanical strength of bioplastics [109].

Piyada et al. [98] have reported a study on rice starch film with rice starch nanocrystals as a reinforcing material and and sorbitol as a plasticizer. As the reinforcing material composition increases, the flexibility of the bioplastic decreases, and the tensile strength increases. Bioplastic without reinforcing material has an elongation at a break of 53.46% and a tensile strength of 7.12 MPa. The use of 30% rice starch nanoparticle as a reinforcing material produces bioplastics with a flexibility of 2.48% and a tensile strength of 12.86 MPa. Another research [132] showed a similar trend, where the reinforcing material composition increased, the flexibility of the bioplastic decreased, and the tensile strength increased. Furthermore, addition of 10% starch nanoparticle as a reinforcing material decreased elongation at break from 1550% to 905% and increased tensile strength from 13.5 MPa to 19.9 MPa. The mechanical properties of bioplastics with various compositions are presented in Table 5.

### 5.2. Water Vapor Permeability

The interaction of the reinforcing material and the matrix in bioplastic packaging forms a strong bond so that the water and air entry paths in the matrix molecules become torturous. This makes it difficult for air and water vapor to penetrate through the packaging film [61]. With a higher concentration of reinforcing material, the structure inside the package will become denser and the pores in the package will be smaller, making it difficult for water and air vapor to penetrate the packaging wall. However, there is also a maximum concentration at which the reinforcing material can work optimally [95]. No data have been found regarding the optimal composition for the use of starch nanoparticles as a reinforcement in bioplastics. This might be due to the optimal concentration of starch nanoparticles being highly dependent on the type of matrix.

The use of reinforcing material is effective in reducing water vapor and oxygen permeability. Bioplastic with a polyurethane matrix (PU) and a reinforcing material of corn starch nanoparticles has a water vapor permeability (WVP) of 3.17% without the use of reinforcing material, which decreases to 0.92% with the use of 30% reinforcing material. Oxygen permeability also decreased from 27.5% with 0% reinforcing material to 6.7% with 30% reinforcing material [95]. A previous study [137] showed that the maximum reinforcing material concentration in the bioplastic composition was 20%, where the matrix used was carboxymethyl chitosan and the reinforcing material used was waxy maize starch nanoparticles with a glycerol plasticizer. When the starch nanocrystal content was higher than 30%, they aggregated and the phase separated in the matrix. The water vapor permeability of bioplastic with various compositions is presented in Table 6.

### 5.3. Thermal Properties

Analysis of the glass transition temperature can be carried out by using DSC (differential scanning calorimetry) [19], DMA (dynamic mechanical analysis) [85], or TGA (thermogravimetry analysis) [95]. The parameters tested include glass transition temperature (T_g_), onset temperature (T_o_), melting point (T_m_), and enthalpy change (∆H) [95]. The mechanism of starch nanoparticles as a reinforcing material in increasing thermal stability involves the interaction between the reinforcing material and matrix, forming a barricade that can inhibit the transfer of heat and energy [141].

According to Hakke et al. [95], there is an increase in ∆H along with the addition of the reinforcing material used. The increasing ∆H is related to starch, with nanoparticle size having a wider active site to interact and bind to the packaging matrix, so it takes more energy to break the polymer structure. This theory is in line with another study on bioplastic with a waterborne polyurethane (WPU) matrix and a pea starch nanoparticle reinforcing material [133]. The enthalpy change in bioplastic with 0% reinforcing material is 2.34 J/g, which increases to 12.5 J/g with a 30% concentration of reinforcing material. However, these results differ from the study by Mukurumbira et al. [85], where along with the addition of reinforcing material concentration, there was a decrease in ∆H. Bioplastic with matrix potato starch, 10% amadumbe starch nanoparticle reinforcing material, and glycerol plasticizer had a ∆H of 0.91 J/g, while in the sample without reinforcing material, the observed ∆H was 14.32 J/g. This could be related to the effect of starch nanoparticles on hindering the lateral arrangements of starch chains and the crystallization of starch films [85]. The thermal properties of bioplastics with various compositions of DSC measurements are shown in Table 7.

### 5.4. Biodegradability

The biodegradability test was carried out to determine the time required for bioplastic packaging to decompose in the soil. Biodegradability causes a decrease in molecular weight, due to the activity of microorganisms, such as bacteria, microalgae, and fungi, resulting in molecular reduction [19]. This test can be performed by placing the film sample in a container filled with soil, then observing the decrease in the weight of the film sample along with the length of the test [142]. During the degradation process, relative humidity plays a role in determining the biodegradation rate. Higher relative humidity can increase the growth of microorganisms, causing a higher rate of degradation [106].

The use of starch nanoparticles as a reinforcing material is effective in increasing the biodegradability of bioplastic packaging. A previous study has reported that within 1 week, bioplastic with a starch mung bean matrix and glycerol plasticizer experienced a weight loss of 28.67%. When adding 1% mung bean starch nanoparticle reinforcing material, in the same period, there was a decrease in packaging weight of 83.46%, and when 2–10% reinforcing material was added, the packaging was 100% degraded [143]. The use of corn starch nanoparticle reinforcing material also increases the biodegradability of PCL packaging. With the use of 10% reinforcing material, the observed weight loss reached 17.6 mg/cm^2^, whereas without reinforcing material, the observed weight loss was 7 mg/cm^2^. Then, the observed weight loss was more than 2 times [19]. An increase in biodegradability is due to the use of starch nanoparticles causing a decrease in the molecular mass of the packaging polymer; hence, the degradation process by microorganisms takes place more quickly [19]. According to González Seligra et al. [106], the use of starch nanoparticles causes an increase in the film’s moisture content, thereby creating a more suitable condition for the growth of microorganisms. This condition increases the chance of microorganism attack. During the degradation process, relative humidity plays a role in determining the biodegradation rate. Higher relative humidity can increase the growth of microorganisms, causing a higher rate of degradation [106]. The biodegradation rate of starch-based bioplastic depends on the hydrophilicity of the matrices. It is directly related to the water absorption; the higher the water absorption, the higher biodegradation [144,145]. The biodegradability of bioplastics with various compositions is shown in Table 8.

## 6. Opportunities and Challenges

Improving the properties of bioplastic polymers is a primary challenge of achieving sustainability in the food and packaging industry [146,147,148]. Starch nanoparticles can be utilized to make other kinds of packaging, including edible packaging, active packaging, and smart packaging, in addition to being a reinforcing material in bioplastic packaging. The usage of starch nanoparticles in edible packaging is safe because they are nontoxic [103]. The use of starch nanoparticles in active packaging can maintain the stability of the bioactive compounds contained in the packaging because these compounds are volatile compounds that are sensitive to certain conditions [136]. Fonseca et al. [149] conducted a study related to the stability of the carvacrol compound mixed with potato starch nanofiber. They found that potato starch nanofibercan be a vehicle for carvacrol release in active packaging.

The use of starch nanoparticles as a reinforcing material as well as in the packaging field in general still finds many obstacles in its commercialization business. This is because production costs have not been effective. After all, the number of adaptations of this technology is still very low. On the other hand, reinforcing materials from inorganic materials have been widely applied, so that the production costs are cheaper and can produce bioplastic packaging with good characteristics. This causes the price of bioplastic packaging with reinforcing materials from starch nanoparticles to be expensive, and the option of using reinforcing materials from inorganic materials is preferred [16].

## 7. Conclusions

Research related to bioplastics as an alternative to conventional plastics is increasingly being carried out. This is based on the environmentally friendly nature of bioplastics. However, bioplastics still have limited characteristics, such as low mechanical strength and high water vapor permeability; hence, reinforcing the material’s role becomes important. The use of starch nanoparticle as a reinforcing material in bioplastic decreases the elongation at break and water vapor permeability, and increases tensile strength and biodegradability. In general, starch nanoparticles also increase the thermal stability of bioplastics. The concentration of reinforcing material that exceeds the maximum threshold has an impact on increasing water vapor permeability and decreasing the mechanical and thermal characteristics of bioplastic packaging because starch nanoparticles at excessive concentrations tend to form aggregates.

## 8. Future Research

Starch nanoparticles have a great opportunity to be developed in the packaging field, mainly because of environmental interest. Although starch nanoparticles have many advantages as reinforcing materials, the utilization of starch nanoparticles on an industrial scale is still limited because adaptation to the use of starch nanoparticles as reinforcing materials in packaging is still low, causing production costs to be high and ineffective. Hence, continuous research about nano-reinforced bioplastic is needed.

In bioplastic processing, the dispersion and homogeneity of mixing bioplastic with starch nanoparticles as a reinforcing material are also interesting to study to produce high-quality bioplastic. The modification of starch nanoparticles can also be a promising option to make better bioplastic characteristics. There have been several studies on starch nanoparticle modification, but the number is still very limited, and thus further research needs to be developed.

## Figures and Tables

**Figure 1 polymers-14-04875-f001:**
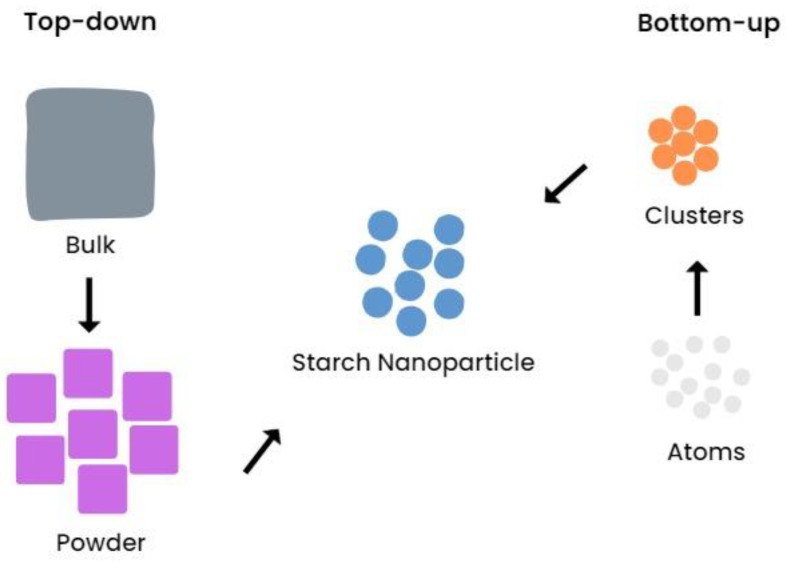
Starch nanoparticle production method with top-down and bottom-up approaches.

**Table 1 polymers-14-04875-t001:** Starch nanoparticle production methods.

Starch Source	Method	Treatment	Results	Ref.
Cassava	Ultrasonication	Time: 30 min Frequency: 20 kHz Temperature: 25 °C	Particle diameter: 35–65 nm Yield: 12%	[59]
Corn	Milling	Media: water Milling speed: 3500 rpm Time: 90 min	Particle size: 245 nm	[47]
Corn	Extrusion	The ratio of starch: water: glycerol = 100:22:23 Storage temperature: 2 °C Storage time: 24 h Extrusion speed: 100–360 rpm Extrusion temperature: 55–110 °C	The higher the extrusion temperature, the smaller the particle size (160 nm)	[57]
Waxy maize	Ultrasonication	Temperature: 8 °C Frequency: 20 kHz Time: 75 min	Particle size: 37 nm	[60]
Cassava	Gamma irradiation	Dose: 20 kGy Speed: 14 kGy/h	Particle size: 31 nm	[54]
Sago	HPH	Media: aquadest Pressure: 250 MPa Time: in 1 h	Particle diameter: 28.514 nm	[61]
Amaranth	AH	Acid: H_2_SO_4_ 3.16 M Temperature: 40 °C Time: 3, 5, and 10 days	Yield on day 3: 17% Particle size: 374 nm	[96]
Potato	EH	Enzyme: α-amylase Incubation time: 30 min Incubation temperature: of 60 °C	Yield: 29% Size: 301 nm	[80]
Potato	AH	Acid: H_2_SO_4_ 3.16 M Homogenization speed: 200 rpm Temperature: 40 °C Time: 5 days	Diameter: 237.03 nm	[83]
Corn	EH	Enzyme: pullulanase Temperature: 58 °C Time: 8 h	Yield: >85%	[79]
Waxy maize	AH	Acid: HCl 2.2 N Temperature: 35 °C Time: 10 days	Size: 30–300 nm	[97]
Elephant foot yam	EH	Enzyme: pullulanase Temperature: 60 °C Time: 8 h	Yield: 61.33% Particle size: 198.14 nm	[84]
Waxy maize	AH	Acid: H_2_SO_4_ 3.16 M Temperature: 40 °C Time: 6 days	Particle size: 47 nm Shape: elliptical	[87]
High-amylose maize			Particle size: 118 nm	
Rice	AH	Acid: H_2_SO_4_ 3 M Temperature: 40 °C Time: 5 days	Crystallinity: 13.3% Crystalline type: A	[98]
Amadumbe	AH	Acid: H_2_SO_4_ 3.16 M Temperature: 40 °C Time: 5 days	Yield: 25% Particle size: 50–100 nm	[85]
Waxy maize	Self-assembly	Enzyme: pullulanase Incubation time: 8 h Incubation temperature: 4 °C	Particle size: 200–300 nm	[99]
High-amylose maize	NP	Solvent: ethanol Temperature: room temperature Time: 8 h	Particle size: 20–80 nm	[82]
Pea			Particle size: 30–150 nm	
Potato			Particle size: 50–225 nm	
Corn			Particle size: 15–80 nm	
Tapioca			Particle size: 30–110 nm	
Sweet potato			Particle size: 40–100 nm	
Waxy maize			Particle size: 20–200 nm	
Corn	AH + ultrasonication	Acid: H_2_SO_4_ 3.16 M Hydrolysis time: 10 days Ultrasonication frequency: 40 kHz Ultrasonication time: 45 min Ultrasonication temperature: 40 °C	Yield: 22% Particle size: 109.9 nm	[100]
Tapioca	NP + sonication	Solvent: ethanol and aquadest Temperature: 22 °C Ultrasonication time: 60 min Ultrasonication frequency of 20 kHz	Particle size: 163 nm	[81]
Waxy maize	Milling + AH	Milling media: ethanol anhydrate Milling speed: 300 rpm Milling time: 15, 30, 45, 60, 75, and 90 min Milling temperature: 40 °C Acid: H_2_SO_4_ 3.16 M Hydrolysis temperature: 40 °C Hydrolysis time: 5 days	The longer the hydrolysis time, the decrease in extraction yield and a reduction in particle size	[101]
Potato	NP	Solution: NaOH Tween (non-ionic surfactant) Solvent: ethanol Solution pH: 7	Diameter: 71.81 nm	[83]
Mango kernel	AH + sonication	Acid: HCl 3.16 M and H_3_PO_4_ 3.16 M Incubation temperature: 40 °C Incubation time: 5 days Sonication was performed within 20 min with frequency of 450 W	Yield: 24.4% Particle size: 24.4 nm	[93]
Mango kernel	AH + sonication	Acid: HCl 3.16 M and H_3_PO_4_ 3.16 M Incubation temperature: 40 °C Incubation time: 5 days Sonication was performed within 20 min with frequency of 450 W	Yield: 31.7% Particle size: 79 nm	[94]
Corn			Yield: 19.4% Particle size: 61.1 nm	

AH: acid hydrolysis; EH: enzymatic hydrolysis: HPH: high-pressure homogenization; H_2_SO_4_: sulfuric acid, HCl: hydrochloric acid, H_3_PO_4_: phosphoric acid, NaOH: sodium hydroxide; NP: nanoprecipitation.

**Table 2 polymers-14-04875-t002:** Granule morphology shape and size of starch nanoparticles from various starch sources and production methods.

Starch Source	Method	Granule Morphology Shape	Particle Size (nm)	Ref.
Banana	NP	Irregular	135	[103]
Waxy maize	Milling + AH	Round to irregular	67.2	[101]
Potato	AH	Elliptic–polyhedric	237.03	[83]
	AH + ultrasonication		153.63	
	NP		71.81	
Waxy rice	AH	Irregular	20–420	[97]
Waxy maize	Cold plasma + ultrasonication	Round and polyhedral	342	[104]
Potato			336	
Tapioca	NP + sonication	Round	163	[81]
	NP	Round	219	
Corn	AH + ultrasonication	Grape-like and parallelepiped	83.9	[100]
Corn	Ultrasonication	Round	36–68	[59]
Cassava			35–65	
Yam			8–32	
Amaranth	AH	Parallelepiped	374	[96]
Waxy maize			322	
Potato	Ultrasonication	Round	74.8	[105]
Waxy maize	Self-assembly	Spherical	200–300	[99]
Corn	Ultrasonication	Round irregular	82	[60]
Waxy maize			297	
Waxy maize	AH	Spherical	58	[34]
	Ultrasonication	Ellipsoidal	37	
Amadumbe	AH	Parallelepiped	50–100	[85]
Mango kernel	AH + sonication	Round	67.1	[93]
Cassava	Gamma irradiation	NR	31	[54]
Waxy maize		NR	41	
Mango kernel	AH + sonication	Spherical	79	[94]
Corn		Spherical	61.1	
Cassava	Gamma irradiation	NR	50–100	[106]
High-amylose maize	NP	Round and oval	20–80	[82]
Pea			30–150	
Potato			50–225	
Corn			15–80	
Tapioca			30–110	
Sweet potato			40–100	
Waxy maize			20–200	
Potato	Self-assembly	Round to irregular	9–40	[107]
Waxy maize			50–120	

AH: acid hydrolysis; NP: nanoprecipitation; NR: not reported.

**Table 3 polymers-14-04875-t003:** The relative crystallinity and crystallinity patterns of starch nanoparticles from various starch sources and production methods.

Starch Source	Method	Relative Crystallinity (%)	Crystalline Type	Ref.
Plantain	AH	90	Type-B	[103]
Waxy maize	Milling + AH	49	NR	[101]
Potato	AH	42.2	Type-B	[83]
	AH + ultrasonication	61.3	Type-B	
	NP	44.1	Type-V	
Potato Maize Cassava	EH	17.5 21.3 13.2	NR NR NR	[80]
Waxy rice	AH	NR	Type-A	[97]
Waxy maize	Cold plasma + ultrasonication	43	Type-A	[104]
Potato		29.1	Type-B	
Tapioca	NP + sonication	15.21	Type-V	[81]
	NP	12.53	Type-V	
Corn	AH + ultrasonication	36.6	NR	[100]
Corn	Ultrasonication	8	Type-A	[59]
Cassava		0	Type-V	
Yam		9	Type-B	
Amaranth	AH	35	Type-A	[96]
Waxy maize		36.5	Type-A	
Potato	Ultrasonication	NR	Type-V	[105]
Waxy maize	EH	55.41	Type B+V	[79]
Corn	Ultrasonication	NR	Type-V	[60]
Waxy maize		NR	Type-V	
Waxy maize	AH	69	Type-A	[34]
	Ultrasonication	0	NR	
Rice	AH	13.3	Type-A	[98]
Amadumbe	AH	NR	Type-A	[85]
Mango kernel	AH + sonication	62.2	Type-A	[93]
Mango kernel	AH + sonication	NR	Type-A	[94]
Corn		NR	Type-A	
Taro	EH	NR	Type B+V	[109]
High-amylose maize	NP	39.8	Type-V	[82]
Pea		31.5		
Potato		26.3		
Corn		23.2		
Tapioca		19.3		
Sweet potato		20.7		
Waxy maize		7.1		
Potato	Self-assembly	54.31	Type-B	[107]
Waxy maize		55.41		

AH: acid hydrolysis; EH: enzymatic hydrolysis; NP: nanoprecipitation; NR: not reported.

**Table 4 polymers-14-04875-t004:** Thermal properties of starch nanoparticles from various starch sources and production methods.

Starch Source	Method	T_o_ (°C)	T_p_ (°C)	T_c_ (°C)	T_c_−T_o_ (°C)	AH (J/g)	Ref.
Plantain	AH	74.6	93.6	106.9	32.3	24.6	[103]
Waxy maize	Cold plasma + ultrasonication	63.18	68.95	77.54	14.36	14.13	[104]
Potato		53.24	56.98	61.95	8.71	12.83	
Tapioca	NP + sonication	52.88	60.15	70.42	23.03	6.61	[81]
	NP	40.77	55.14	67.08	20.7	4.64	
Waxy rice	AH	62.68	68.62	77.95	15.27	3.3	[97]
High-amylose maize	NP	48.19	61.20	71.20	23.01	6.16	[82]
Pea		46.51	60.42	73.47	26.96	4.57	
Potato		43.45	66.42	73.31	29.86	4.34	
Corn		41.21	55.29	70.31	29.10	4.21	
Tapioca		44.21	54.14	67.62	23.41	2.37	
Sweet potato		42.74	56.24	70.21	27.47	3.02	
Waxy maize		44.13	59.54	70.44	26.31	0.96	
Potato	Self-assembly	65.97	97.39	101.32	35.35	−5.34	[107]
Waxy maize		62.4	78.31	93.24	30.84	−10.17	
Waxy maize	AH	57.70	92.40	81.90	24.20	34.6	[63]
Corn		60.20	89.10	116.70	56.50	20.20	

AH: acid hydrolysis; NP: nanoprecipitation.

**Table 5 polymers-14-04875-t005:** Mechanical properties of bioplastics with various compositions.

Bioplastic Composition			Elongation at Break (%E)	Young Modulus (MPa)	Tensile Strength (MPa)	Ref.
Matrix	Reinforcing Material	Plasticizer				
Polyurethane	Corn SNP 0% 5% 10% 20% 30%	-	29.8 44.6 67 61.1 70.3	NR	8.5 11.8 15.9 16.2 17.1	[95]
Cassava starch (4)	Cassava SNP (0) (0.5) (5) (10)	Glycerol (2.1)	NR	4.8 5 15.2 29.8	1 1.24 1.98 3.15	[19]
Sago starch	Sago SNP 0% 2% 4% 6% 8%	-	1.24 1.5 1.62 1.67 1.58	2.314 2.87 3.21 2.532 2.444	2.469 2.902 3.578 2.669 2.546	[61]
Waterborne polyurethane	Pea SNP 0% 5% 10% 20% 30%	-	825 718 600 526 300	3 7 115 195 208	11.5 29 31 25 14	[133]
Polycaprolactone	Corn SNP 0% 2.25% 5% 10%	-	1550 1455 1410 905	274.8 310.7 333.6 339.3	13.5 17.7 19.5 19.9	[132]
Rice starch	Rice SNP 0% 5% 10% 15% 20% 25% 30%	Sorbitol 40%	53.46 34.76 28.75 17.12 8.89 2.51 2.48	NR	7.12 10.82 11.53 12.79 16.43 13.91 12.86	[98]
Potato starch	Amadumbe SNP 0% 2.5% 5% 10%	Glycerol	NR	NR	2.09 8.11 5.91 6.7	[85]
Amadumbe starch	Amadumbe SNP 0% 2.5% 5% 10%	Glycerol	NR	NR	2.4 3.89 3.37 2.08	
Cross-linked cassava starch	Cassava SNP 0% 2% 4% 6% 8%	Glycerol	175.87 138.21 98.85 76.46 60	10 10.98 11.87 12.96 12.5	5.48 7 8.97 9.17 8.75	[134]
Corn starch	Corn SNP 0 2.5% 5% 7.5% 10%	Glycerol	770.5 746.3 997 1112.9 1462	10.95 9.32 11.95 12.12 15.87	20.34 20.98 9.86 9.35 2.85	[94]
Mango kernel starch	Mango kernel SNP 0% 2.5% 5% 7.5% 10%	Glycerol	659.1 793.8 1466.2 1899.4 1437	9.3 10.39 17.51 18.96 15.47	22.85 19.61 11.15 1.53 2.03	
Pea starch	Potato SNP 0% 3% 6% 9% 12%	Glycerol 3 g	53.4 52.7 48.6 45 37	NR	8.8 11.5 15 9.8 9.5	[99]
Corn starch	Taro SNP 0% 0.5% 2% 5% 10% 15%	Glycerol 3 g	84.5 80 74.7 66.8 64.1 58	NR	1.1 1.53 1.74 2.51 2.87 2.28	[109]
Pea starch	Waxy maize SNP 0% 1% 3% 5% 7% 9%	Glycerol 3%	29.23 26.18 20.46 12.58 21.6 26.7	21.15 27.95 37.89 85.72 36.59 27.56	5.76 6.56 6.95 9.96 7.12 6.68	[135]
Soy protein isolate 0.25 g	Corn SNP 0% 2% 5% 10% 20% 40%	Glycerol 0.125 g	65.95 53.79 58.67 32.17 41.89 21.35	26.89 55.31 39.42 71.05 102.23 310.34	1.1 1.42 1.34 1.79 2.61 5.08	[136]
Carboxymethyl chitosan 5 g	Waxy maize SNP 0% 3% 6% 10% 15% 20% 30% 40%	Glycerol 2 g	180.24 180 160 148.79 137.68 115.47 97.36 62	NR	15.36 17 18.78 19.94 21.5 26.87 28.32 26.9	[137]
Pullulan	Waxy maize SNP 0% 3% 6% 10% 15% 20% 30% 40%	Sorbitol 30%	237 139 120 110 67 58 40 16	94 100 124 586 687 700 1004 1295	6 8 9.3 9.5 14 15.2 19.8 26.2	[138]
Waxy maize starch	Waxy maize SNP 0% 5% 10% 15%	Sorbitol 25%	63 57 58 41	17.2 36.6 38.3 46.2	0.38 0.99 1.37 1.59	[139]

NR: not reported; SNP: starch nanoparticle.

**Table 6 polymers-14-04875-t006:** Water vapor permeability of bioplastic with various compositions.

Bioplastic Composition			WVP (g/ Pa.m.h)	Ref.
Matrix	Reinforcing Material	Plasticizer		
Sago starch	Sago SNP 0% 2% 4% 6% 8%	-	12.08 × 10^−3^ 7.80 × 10^−3^ 6.81 × 10^−3^ 5.93 × 10^−3^ 6.42 × 10^−3^	[61]
Starch rice	SNP rice 0% 5% 10% 15% 20%	Sorbitol 40%	0.75 × 10^−13^ 0.66 × 10^−13^ 0.60 × 10^−13^ 0.31 × 10^−13^ 0.30 × 10^−13^	[98]
Potato starch	Amadumbe SNP 0% 2.5% 5% 10%	Glycerol	1.8 × 10^−5^ 1.6 × 10^−5^ 1.5 × 10^−5^ 1.5 × 10^−5^	[85]
Amadumbe starch	Amadumbe SNP 0% 2.5% 5% 10%	Glycerol	2.3 × 10^−5^ 2.1 × 10^−5^ 2.0 × 10^−5^ 1.8 × 10^−5^	
Waxy maize starch	Waxy maize SNP 0% 2.5%	Glycerol 33%	1.37 × 10^−6^ 2.45 × 10^−6^	[140]
Corn starch	Corn SNP 0% 2.5% 5% 7.5% 10%	Glycerol	1.41 × 10^−6^ 1.25 × 10^−6^ 1.15 × 10^−6^ 1.11 × 10^−6^ 1.12 × 10^−6^	[94]
Mango kernel starch	Mango kernel SNP 0% 2.5% 5% 7.5% 10%	Glycerol	1.37 × 10^−6^ 1.35 × 10^−6^ 1.21 × 10^−6^ 1.08 × 10^−6^ 1.00 × 10^−6^	
Corn starch 7.5 g	Taro SNP 0% 0.5% 2% 5% 10% 15%	Glycerol 3 g	2.74 × 10^−7^ 2.05 × 10^−7^ 1.83 × 10^−7^ 1.49 × 10^−7^ 1.20 × 10^−7^ 1.37 × 10^−7^	[109]
Pea starch 5 g	Waxy maize SNP 0% 1% 3% 5% 7% 9%	Glycerol 1.5 g	11.18 × 10^−3^ 7.57 × 10^−3^ 6.09 × 10^−3^ 4.26 × 10^−3^ 5.41 × 10^−3^ 5.50 × 10^−3^	[135]
Soy protein isolate 0.25 g	Corn SNP 0% 5% 20% 40%	Glycerol 0.125 g	4.3 × 10^−6^ 4.8 × 10^−6^ 3.9 × 10^−6^ 3.57 × 10^−6^	[136]
Cassava starch 10 g	Waxy maize SNP 0% 2.5%	Glycerol 5 g	1.62 × 10^−6^ 0.97 × 10^−6^	[131]

SNP: starch nanoparticle.

**Table 7 polymers-14-04875-t007:** Thermal properties of bioplastics with various compositions.

Bioplastic Composition			T_g_ (°C)	T_m_ (°C)	T_o_ (°C)	∆H (J/g)	Ref.
Matrix	Reinforcing Material	Plasticizer					
Potato starch	Amadumbe SNP 0% 2.5% 5% 10%	Glycerol 30%	64 72 94 94	78.19 88.59 107.31 105.52	78.19 88.59 107.31 105.52	14.32 16.69 2.39 0.91	[85]
Amadumbe starch	Amadumbe SNP 0% 2.5% 5% 10%	Glycerol 30%	60 70 86 92	66.03 72.49 73.07 96.24	6603 72.49 73.07 96.24	24.92 20.68 15.63 15.27	
Corn starch 7.5 g	Taro SNP 0% 0.5% 2% 5% 10% 15%	Glycerol 3 g	NR	210.21 219.47 223.19 218.1 222.34 209.91	171.75 187.59 187.43 184.08 187.28 180.3	45.49 51.9 50.84 50.03 55.34 36.95	[109]
Pea starch 7.5 g	Potato SNP 0% 3% 6% 9% 12%	Glycerol 3 g	NR	225.81 226.91 227.2 231.98 235.81	192.97 195.01 196.43 199.58 200.08	42.75 24.72 22.24 32.51 34.32	[107]
Pea starch 5 g	Waxy maize SNP 0 1% 3% 5% 7% 9%	Glycerol 3%	NR	188.51 190.81 189.56 193.92 189.66 190.1	186.61 186.26 186.79 187.35 186.61 187.54	23.51 24.34 24.75 26.76 22.31 22.17	[135]
Waxy maize starch	Waxy maize SNP 0% 5% 10% 15%	Sorbitol 25%	40.1 46.3 52.3 58.8	150.1 152.7 160.4 169.7	NR	99.8 122.4 150.7 165.2	[139]

NR: not reported; SNP: starch nanoparticle.

**Table 8 polymers-14-04875-t008:** Biodegradability of bioplastics with various compositions.

Bioplastic Composition			Duration	Weight Loss	Ref.
Matrix	Reinforcing Material	Plasticizer			
Cassava starch (4)	Cassava SNP (0) (0.5) (5) (10)	Glycerol (2.1)	17 weeks	82.8% 82.4% 82.5% 84.7%	[19]
Polycaprolactone	Corn SNP 0 2.25 5 10	-	4 days	7 mg/cm^2^ 10 mg/cm^2^ 15.7 mg/cm^2^ 17.6 mg/cm^2^	[132]
Pea starch	Pea SNP 0% 0.5% 1% 2% 5% 10%	Glycerol 50%	1 week	28.67% 49.25% 83.46% 100% 100% 100%	[143]

SNP: starch nanoparticle.

## Data Availability

Not applicable.

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
