# Peer review of "A Comprehensive Study on Starch Nanoparticle Potential as a Reinforcing Material in Bioplastic"

_polymers, 2022, doi:10.3390/polym14224875_

Round 1

Reviewer 1 Report

L61 Please separate the sentence.

Section 2.1 There should be discussion about method and properties.

Table 5 -> What is gr? gram? Should they be % to be consistent to other data in the Table?

Table 6 -> Add unit for WVP

The WVP values in this Table differ highly among study (x 10^-5 to 10^-15). Please recheck unit. Add discussion about this difference.

Table 7 -> Please confirm that these Tg values came from DSC measurement.

Discuss large difference among Tg values-46 to 94 C.

L362 Add more discussion e.g., Biodegradation rate of starch-based materials depended on hydrophilicity of the matrices. Higher water absorption accelerated biodegradation (Wadaugsorn 2022 Industrial Crops and Products; Wongphan 2022 Food Packaging and Shelf Life).

L369 Add more discussion e.g., Improving properties of bioplastic polymers are primary challenge to achieve sustainability in food and packaging industry (Promhuad 2022 Polymers, San 2022 Polymers, Srisa 2022 Polymers).

Conclusions -> What about future research on processing e.g., dispersion and homogeneity mixing, etc.

Author Response

Thank you very much for your comments concerning our manuscript entitled “A Comprehensive Study on Starch Nanoparticle Potential as Reinforcing Material in Bioplastic”. Those comments are valuable and very helpful for revising and improving our paper. We have studied the comments carefully and have made a correction which we hope meets with approval. The revised portions are marked in red on the paper. The main correction and the responses to the reviewer’s comment are as follows:

Reviewer 1

L61 Please separate the sentence.

Response:

It has been revised (Line 68-73)

Section 2.1 There should be discussion about method and properties.

Response:

Table 5 -> What is gr? gram? Should they be % to be consistent to other data in the Table?

Response:

Gr has been changed to g

It has been converted from gr to %

Table 6 -> Add unit for WVP

Response:

WVP unit has been added

The WVP values in this Table differ highly among study (x 10^-5 to 10^-15). Please recheck unit. Add discussion about this difference.

Response:

Thank you for your suggestions. We have rechecked the units and we found an error in converting the units. We have corrected the error and made it in units of g/Pa.m.h. (Table 6)

Table 7 -> Please confirm that these Tg values came from DSC measurement. Discuss large difference among Tg values -46 to 94 C.

Response:

Thank you for your comment. We are sorry. Some of data cannot compare each other because they have different methods to measure the Tg value. The irrelevant data have been deleted from the Table 7.

L362 Add more discussion e.g., Biodegradation rate of starch-based materials depended on hydrophilicity of the matrices. Higher water absorption accelerated biodegradation (Wadaugsorn 2022 Industrial Crops and Products; Wongphan 2022 Food Packaging and Shelf Life).

Response:

It has been added (Line 431-434)

L369 Add more discussion e.g., Improving properties of bioplastic polymers are primary challenge to achieve sustainability in food and packaging industry (Promhuad 2022 Polymers, San 2022 Polymers, Srisa 2022 Polymers).

Response:

It has been added (Line 441-442)

Conclusions -> What about future research on processing e.g., dispersion and homogeneity mixing, etc.

Response:

It has been added (section 8. Future research line 473-484)

Reviewer 2 Report

The manuscript “A Comprehensive Study on Starch Nanoparticle Potential as 2 Reinforcing Material in Bioplastic” is interesting, aiming at review of nanoparticle preparation, nanoparticle property, bioplastic application, performance characterization, challenges, etc. However, there are some problems needing modification. Comments:

(1) Abstract needs to be rephrased logistically.

(2) Line 49, what role of amylose and amylopectin played in starch application?

(3) Line 85, “The physical method tends to result in starch nanoparticles with low crystallinity”, why? Mechanism? References?

(4) Lines 109-110, “Although acid hydrolysis can produce high crystallinity starch nanoparticles, the time required for the production is far longer than enzymatic hydrolysis and nanoprecipitation”, Why? Mechanism? Crystallinity comparison references?

(5) Lines 114-116, “Acid hydrolysis produces starch nanoparticles with higher crystallinity, compared to enzymatic hydrolysis, but, enzymatic hydrolysis tends to produce a higher yield of the starch nanoparticle.” Please provide the crystallinity comparison and offer references. Also, in Lines 116-120, the differences in yield may be attributed to starch type in addition to hydrolysis types.

(6) what’s the mechanism underlining nanoprecipitation?

(7) Line 122-123, why? Mechanism? References?

(8) Line 140-142 refers the overcoming of long treatment time. However, Line 142-145 does not refer the treatment time. Please improve the logistic in this part and the similar problems in the manuscript.

(9) Line 165-173 is confusing. Please specify the relationship between amylose content and nanoparticles, the relationship between amylose content and preparation method, the relationship between amylose content and crystallinity, and the relationship between amylose content and particle size.

(10) Line 180-182, “a previous study”? Please provide the references.

(11) What’s the mechanism of different crystallinity underling the different preparation methods (Table 3)? Please summarize it carefully.

(12) Line 251-253, “Subsequently, the mixture is cooled down and starch nanoparticles (reinforc- 251 ing material) were added. Cooling down before adding starch nanoparticles is addressed 252 to hinder the gelatinization of the starch nanoparticle.” What mixture? Also, please improve the sentences for better reading.

(13) Line 260-262, do you mean that starch nanoparticles are non-beneficial for producing bioplastic?

(14) Line 309-310, what’s the suitable concentration range?

(15) Line 312, “PU”, what’s it?

(16) Line 313-314, “the figure drops to 0.92%” what does it mean?

(17) Line 332-342, what’s the possible reasons for the differences?

(18) Line 353-354, why? Mechanism?

(19) Line 388-393, please use more professional sentences.

(20) Line 397-400, more contents with expertise should be summarized.

(21) Several figures are needed for better reading.

(22) Grammar, sentences and small writing mistakes should be carefully checked, for example Line 126.

Author Response

Thank you very much for your comments concerning our manuscript entitled “A Comprehensive Study on Starch Nanoparticle Potential as Reinforcing Material in Bioplastic”. Those comments are valuable and very helpful for revising and improving our paper. We have studied the comments carefully and have made a correction which we hope meets with approval. The revised portions are marked in red on the paper. The main correction and the responses to the reviewer’s comment are as follows:

Reviewer 2

The manuscript “A Comprehensive Study on Starch Nanoparticle Potential as 2 Reinforcing Material in Bioplastic” is interesting, aiming at review of nanoparticle preparation, nanoparticle property, bioplastic application, performance characterization, challenges, etc. However, there are some problems needing modification. Comments:

  • Abstract needs to be rephrased logistically.

Response:

The sentences have been rearranged [Line 13-25]

  • Line 49, what role of amylose and amylopectin played in starch application?

Response:

It has been added (line 52-57)

  • Line 85, “The physical method tends to result in starch nanoparticles with low crystallinity”, why? Mechanism? References?

Response:

It has been revised (line 94-98)

  • Lines 109-110, “Although acid hydrolysis can produce high crystallinity starch nanoparticles, the time required for the production is far longer than enzymatic hydrolysis and nanoprecipitation”, Why? Mechanism? Crystallinity comparison references?

Response:

It has been added (line 125-130, 135-143)

  • Lines 114-116, “Acid hydrolysis produces starch nanoparticles with higher crystallinity, compared to enzymatic hydrolysis, but, enzymatic hydrolysis tends to produce a higher yield of the starch nanoparticle.” Please provide the crystallinity comparison and offer references. Also, in Lines 116-120, the differences in yield may be attributed to starch type in addition to hydrolysis types.

Response:

It has been added (line 139-143, 149-150)

  • what’s the mechanism underlining nanoprecipitation?

Response:

It has been added (line 125-130)

  • Line 122-123, why? Mechanism? References?

Response:

The statement has been added (line 151-157)

  • Line 140-142 refers the overcoming of long treatment time. However, Line 142-145 does not refer the treatment time. Please improve the logistic in this part and the similar problems in the manuscript.

Response:

It has been revised (line 177-182)

  • Line 165-173 is confusing. Please specify the relationship between amylose content and nanoparticles, the relationship between amylose content and preparation method, the relationship between amylose content and crystallinity, and the relationship between amylose content and particle size.

Response:

It has been revised (line 211-219)

  • Line 180-182, “a previous study”? Please provide the references.

Response:

It has been added (line 227)

  • What’s the mechanism of different crystallinity underling the different preparation methods (Table 3)? Please summarize it carefully.

Response:

It has been added (line 257-261)

  • Line 251-253, “Subsequently, the mixture is cooled down and starch nanoparticles (reinforcing material) were added. Cooling down before adding starch nanoparticles is addressed 252 to hinder the gelatinization of the starch nanoparticle.” What mixture? Also, please improve the sentences for better reading.

Response:

It has been revised (line 302-303)

  • Line 260-262, do you mean that starch nanoparticles are non-beneficial for producing bioplastic?

Response:

Sentences has been rearranged (line 312-313)

  • Line 309-310, what’s the suitable concentration range?

Response:

It has been added (360-363)

  • Line 312, “PU”, what’s it?

Response:

It has been added (Line 365)

  • Line 313-314, “the figure drops to 0.92%” what does it mean?

Response:

It has been revised (Line 367)

  • Line 332-342, what’s the possible reasons for the differences?

Response:

It has been added (386-389, 397-398)

  • Line 353-354, why? Mechanism?

Response:

It has been added (Line 423-434)

  • Line 388-393, please use more professional sentences.

Response:

This paragraph has been rearranged and some points were added (line 465-471).

  • Line 397-400, more contents with expertise should be summarized.

Response:

It has been added (Line 473-484)

  • Several figures are needed for better reading.

Response:

It has been added (Figure 1)

  • Grammar, sentences and small writing mistakes should be carefully checked, for example Line 126.

Response:

It has been revised (line 151-156)

Reviewer 3 Report

The paper "A Comprehensive Study on Starch Nanoparticle Potential as Reinforcing Material in Bioplastic" is interesting and may be published after minor corrections;

(a) Some images on the topic addressed can be considered by the authors, note that in this type of review article, the placement of figures is an interesting factor to consider;

(b) Some outstanding research on the topic, or on peripheral issues should be considered by the authors, such as: 10.3390/ma14133549; 10.1016/j.cscm.2015.12.003; 10.1016/j.cscm.2019.e00319;

(c) Some tables are very extensive, authors should be more concise and objective;

(d) A topic on future perspective and gaps should be inserted at the end of the paper. This should be broken down from the conclusion (item 7), as discussed.

Author Response

Thank you very much for your comments concerning our manuscript entitled “A Comprehensive Study on Starch Nanoparticle Potential as Reinforcing Material in Bioplastic”. Those comments are valuable and very helpful for revising and improving our paper. We have studied the comments carefully and have made a correction which we hope meets with approval. The revised portions are marked in red on the paper. The main correction and the responses to the reviewer’s comment are as follows:

Reviewer 3

The paper "A Comprehensive Study on Starch Nanoparticle Potential as Reinforcing Material in Bioplastic" is interesting and may be published after minor corrections;

  • Some images on the topic addressed can be considered by the authors, note that in this type of review article, the placement of figures is an interesting factor to consider;

Response:

The figure has been added (Figure 1)

  • Some outstanding research on the topic, or on peripheral issues should be considered by the authors, such as: 10.3390/ma14133549; 10.1016/j.cscm.2015.12.003; 10.1016/j.cscm.2019.e00319;

Response:

It has been added (Line 31 and 33)

  • Some tables are very extensive, authors should be more concise and objective;

Response:

It has been revised

  • In Table 7, the irrelevant data have been deleted. Some of the data cannot compare to each other because they have different methods to measure the Tg value.
  • A topic on future perspectives and gaps should be inserted at the end of the paper. This should be broken down from the conclusion (item 7), as discussed.

Response:

          It has been revised (Section 8. Future research)

Round 2

Reviewer 1 Report

The manuscript has been improved.

Reviewer 2 Report

Accept